**Data Availability Statement:** All relevant data are within the manuscript and its Supporting Information files.

**Funding:** This work was supported by grants from the Ministry of Science and Technology in Taiwan

# Pathway-targeting gene matrix for Drosophila gene set enrichment analysis

**Jack Cheng**[1,2☯], **Lee-Fen Hsu**[3,4,5☯], **Ying-Hsu Juan**[1☯], **Hsin-Ping Liu**[6]*, **Wei-Yong Lin**[1,2,7]*

**1** Graduate Institute of Integrated Medicine, China Medical University, Taichung, Taiwan, **2** Department of Medical Research, China Medical University Hospital, Taichung, Taiwan, **3** Department of Respiratory Care, Chang Gung University of Science and Technology, Puzi City, Chiayi County, Taiwan, **4** Division of Neurosurgery, Department of Surgery, Chang Gung Memorial Hospital, Puzi City, Chiayi County, Taiwan, **5** Chronic Disease and Health Promotion Research Center, Chang Gung University of Science and Technology, Puzi City, Chiayi County, Taiwan, **6** Graduate Institute of Acupuncture Science, China Medical University, Taichung, Taiwan, **7** Brain Diseases Research Center, China Medical University, Taichung, Taiwan

☯ These authors contributed equally to this work.

* hpliu@mail.cmu.edu.tw (HPL); linwy@mail.cmu.edu.tw (WYL)

## Abstract

Gene Set Enrichment Analysis (GSEA) is a powerful algorithm to determine biased pathways between groups based on expression profiling. However, for fruit fly, a popular animal model, gene matrixes for GSEA are unavailable. This study provides the pathway-targeting gene matrixes based on Reactome and KEGG database for fruit fly. An expression profiling containing neurons or glia of fruit fly was used to validate the feasibility of the generated gene matrixes. We validated the gene matrixes and identified characteristic neuronal and glial pathways, including mRNA splicing and endocytosis. In conclusion, we generated and validated the feasibility of Reactome and KEGG gene matrix files, which may benefit future profiling studies using Drosophila.

## Introduction

Gene Set Enrichment Analysis (GSEA) is an algorithm [1] that determines whether a previously defined set of genes shows significant differences between two groups of biological samples. Since 2005, GSEA has been widely applied in profiling studies with more than 20,000 citations. In contrast to conventional fold-change (FC) ranking methods, GSEA does not require a manually defined cutoff, e.g., FC > 2, but determines whether members of a gene set tend to occupy the top (or bottom) of the FC list. Therefore, GSEA may provide neglected information due to the cutoff bias in the conventional FC ranking methods.

To apply GSEA, in addition to the profiling data, a Gene Matrix file (e.g., *.gmt) or its alternatives, describing the constitution of gene sets is required. Although 28,705 *Homo sapiens* gene sets are available in the Molecular Signatures Database (MSigDB) (http://www.gsea-msigdb.org/gsea/msigdb/collections.jsp), gene sets for other organisms are largely limited (http://ge-lab.org/gskb/) and majorly generated from Gene Ontology data.

(MOST108-2320-B-039-031-MY3 to HPL, MOST 109-2314-B-039-030 and MOST 110-2314-B-039-009 to WYL), form Chang Gung Memorial Hospital (CMRPF6H009 and CMRPF6L0011 to LFH), and from China Medical University & Hospital (CMU109-MF-85, CMU108-MF-68, and DMR-109-150 to WYL, CMU108-MF-61 to HPL). The funders had no role in this study. MOST Taiwan:www. most.gov.tw Chang Gung Memorial Hospital: www.cgmh.org.tw China Medical University: www. cmu.edu.tw China Medical University Hospital: www.cmuh.cmu.edu.tw.

**Competing interests:** The authors have declared that no competing interests exist.

Gene Ontology is a structured, precisely defined, controlled vocabulary for describing the roles of genes with three independent ontologies, i.e., biological process, molecular function, and cellular component [2]. Briefly, biological process indicates the biological objective of the gene/protein; molecular function describes its biochemical activity, while cellular component refers to its subcellular distribution. Although GO allows an easy and quick understanding of the roles of a gene, however, "it describes only what is done without specifying where or when the event actually occurs." as stated in the original GO paper [2]. This knowledge gap is exactly what KEGG [3] and Reactome [4] try to fill. Both databases provide sequential information and partnership of the reaction of the gene/protein. On the contrary, the dependence on the published/curated scientific literature largely limits the application of KEGG/Reactome on genes of unknown function, while GO may cover this part by similarity prediction.

Thus, the choice of gene sets in GSEA is largely dependent on the purpose of the study. For example, two previous GSEA papers [5, 6] may be improved by adopting KEGG/Reactome gene sets to provide more details of the affected pathways, while for another study [7], GO gene sets is perfect for its goal of predicting functions of unknown genes.

As an experimental animal model, the fruit fly (*Drosophila melanogaster*) is a powerful tool to decipher genetic mechanisms both in behaviors and human diseases [8], owing to the convenience in gene-manipulating [9]. Furthermore, Reactome and KEGG are both manually curated and peer-reviewed pathway databases. Therefore, the objective of this study is to provide the pathway-targeting gene matrix files based on Reactome and KEGG for fruit fly profiling studies using GSEA. After generating the gene matrix files, the expression data of two distinct fruit fly cell phenotypes, i.e., neurons and glia, are run in the GSEA software to identify enriched gene sets typical for the respective cell types, which would support the validity of the gene matrix files.

## Method

### Generation of Reactome and KEGG gene matrix files

The curated pathway-gene information was retrieved from the Reactome and KEGG websites. The gene matrix files (S1 and S2 Files) were generated according to the GSEA data formats, i.e., a tab-delimited file, and each row represents a gene set.

Specifically, the Drosophila-specific genes of KEGG pathways were downloaded from the "KEGG Pathway Maps—Drosophila melanogaster (fruit fly)" with the website https://www. genome.jp/brite/query=00190&htext=br08901.keg&option=-a&node_proc=br08901_ org&proc_enabled=dme&panel=collapse. By clicking each "tringle symbol" of mother categories, sub-categories will expand. By clicking the number preceding each sub-category, e.g., 00010 of Glycolysis / Gluconeogenesis, it will bring you to the map of the specific pathway (https://www.genome.jp/kegg-bin/show_pathway?dme00010). Further clicking the title of pathway map on the upper left corner will finally bring you to the detail page of that pathway (https://www.genome.jp/entry/dme00010). At the upper right corner of the page, an "all links" box contains the "KEGG GENES" list. Repeat the process to exhaust the Drosophila KEGG pathways.

The Drosophila-specific genes of Reactome pathways were downloaded from the (https:// reactome.org/PathwayBrowser/#/R-DME-XXXXXXX, where XXXXXXX denotes for seven digits of a specific pathway, e.g., 9612973 for autophagy). There are three panels on the page. The left panel shows the hierarchy of Drosophila pathways in Reactome, while at the right lower panel, by clicking the tab "Molecules", then the "protein" link, the gene/protein list is available. Repeat the process to exhaust the Drosophila Reactome pathways on the left panel.

A gmt file is a tab-separated plain text, and each row describes one gene set. In each row, the first column contains the name of the gene set, while the second column contains additional details, e.g., KEGG ID of the pathway (gene set). The gene set members, i.e., FlyBase IDs in this study, are listed from the third column of the row, one gene in one column. Thus, once the gene list of pathways is available, the gmt file can be generated by locating the pathway elements into corresponding cells with any plain text editor or Microsoft Excel. After saving the file as a tab-separated plain text, modify the filename extension, i.e., *.txt, to *.gmt in the file browser.

## Collection of validation data

The expression profiling of GFP positive cells with Repo or Elav driver sorted from brains of Drosophila of the accession ID GSE45344, provided by DeSalvo MK & Bainton RJ [10], was downloaded from the NCBI GEO database. Elav is a gene encoding an RNA binding protein capable of regulating mRNA processing exclusively expressed in neurons, and Repo encodes a transcription factor specifically expressed in glia. By using GAL4-UAS reporter system, Repo-GAL4 drives the expression of UAS-GFP specifically in glia, while Elav-GAL4 drives UAS-GFP exclusively in neurons. Fluorescence activated cell sorting (FACS) is a technique to separate cells as they flow past stimulating lasers [11]. The downloaded file was saved as a tab-delimited txt file (S3 File) as an input to the GSEA 4.1.0 software. Notably, the probe IDs were converted to Flybase annotation ID (CG_ID) according to the conversion table (S4 File). The annotation of Microarray probe IDs is available from the "SOFT formatted family file" at https://ftp.ncbi.nlm.nih.gov/geo/series/GSE45nnn/GSE45344/soft/. Specifically, from line 110 of the file, the 1st column is the Microarray probe ID, and the 3rd column is the FlyBase annotation ID, i.e., CG_ID. However, the context must be "cleared" to extract the correct FlyBase ID, e.g., for "CG16844-RA", the "-RA" must be trimmed to get the correct ID "CG16844". If the dataset of your interest does not provide the probe ID annotation, you may try the gene ID conversion tool on https://david.abcc.ncifcrf.gov/conversion.jsp or from the website of the gene chip manufacturer. The non-Flybase-symbol-corresponding probes were omitted (e.g., those from cDNA library, pseudogene, or transposon). The intensity values were log2-based, and they were exponentially transformed before enrichment analysis. Only the record with the strangest average intensity was used for the validation, in the case of redundant intensities presented for an identical CG_ID, i.e., the highest value was chosen in case of multiple probe mappings to the same CG_ID.

## Validation of generated gene matrix files

When running the GSEA 4.1.0 software according to the GSEA user guide, S3 File was assigned as the expression dataset, while S1 and S2 Files were assigned as the Gene set database. Notably, the "Collapse/Remap to gene symbols" parameter must be assigned as "No_collapse", which means using the identifiers in the dataset as is in the original format. Phenotype labels were manually inputted by first selecting the "Create an on-the-fly phenotype" pull-down, then typing in the sample ID of the two groups, i.e., Elav_1 and Repo_1, in "Samples for class A" and "Samples for class B", respectively.

## Results

### Characteristics of the generated Reactome and KEGG gene matrix files

The Drosophila Reactome pathways are presently classified into nine categories: circadian clock pathway, Hedgehog pathway, Hippo/Warts pathway, Imd pathway, insulin receptor-

mediated signaling, JAK/STAT pathway, planar cell polarity pathway, Toll pathway, and Wingless pathway. There are currently 1450 pathways annotated and supplied with genetic information. The generated Drosophila Reactome gene matrix file is provided as S1 File.

In contrast, the Drosophila KEGG pathways are presently classified into five categories, including genetic information processing, environmental information processing, cellular processes, organismal systems, and metabolism. Although there are 137 pathways annotated, only 131 of them are currently supplied with genetic information. The generated Drosophila KEGG gene matrix file is provided as S2 File.

## Validation of Reactome gene matrix file

A profiling dataset comparing ELAV-GFP (representing neurons) and REPO-GFP (representing glia) sorting cells from the Drosophila brain was used to test the feasibility of the generated gene matrix files. With the default Gene set size filters (min = 15, max = 500), 835 out of the 1450 gene sets were filtered out, and the remaining 615 gene sets were used in the analysis. All 13615 genes of the profiling dataset were included.

The Reactome gene matrix file worked well, and as a result, 5616 (41.2%) of the genes were associated with phenotype Class A (ELAV group) with a correlation area of 40.7%; while 7999 (58.8%) of the genes were associated with Class B (REPO group) with correlation area 59.3%. The heat map of the top 50 genes (Fig 1A) and the ranked gene list correlation profile (Fig 1B) are shown. The global enrichment scores across gene sets (Fig 1C) show a typical 2-peak separation for ELAV and REPO groups.

For the Class A (ELAV group), under the criterion of false discovery rate (FDR) below 25%, 214 or 148 gene sets were upregulated significantly at a nominal p-value of 5% or 1%, respectively. Representative enrichment plots (Fig 2) highlighted mRNA splicing, transport of mature transcript to cytoplasm, synthesis of PIPs at the plasma membrane, and RAC1 GTPase cycle. For the Class B (REPO group), under the criterion of false discovery rate (FDR) below 25%, 68 or 52 gene sets were upregulated significantly at a nominal p-value of 5% or 1%, respectively. Representative enrichment plots (Fig 3) highlighted peptide hormone metabolism, mitochondrial fatty acid beta-oxidation, assembly of active LPL & LIPC lipase complexes, and activation of matrix metalloproteinase. The detailed Reactome enrichment results for Class A & B are provided as S5 and S6 Files, respectively.

## Validation of KEGG gene matrix file

The same dataset was also used for testing the feasibility of the generated KEGG gene matrix file. The KEGG gene matrix file also worked well. With the default Gene set size filters, 33 out of the 131 gene sets were filtered out, and the remaining 98 gene sets were used in the analysis. The global enrichment scores across gene sets (S1 File) also shows a typical 2-peak separation for ELAV and REPO groups. For the Class A (ELAV group), under the criterion of false discovery rate (FDR) below 25%, 27 or 20 gene sets were upregulated significantly at a nominal p-value of 5% or 1%, respectively. Representative enrichment plots (Fig 4) highlighted spliceosome, endocytosis, WNT signaling, and mTOR signaling pathway. For the Class B (REPO group), under the criterion of false discovery rate (FDR) below 25%, 31 or 19 gene sets were upregulated significantly at a nominal p-value of 5% or 1%, respectively. Representative enrichment plots (Fig 5) highlighted fatty acid degradation, glutathione metabolism, ABC transporters, and glycine, serine, & threonine metabolism. The detailed KEGG enrichment results for Class A & B are provided as S7 and S8 Files, respectively.

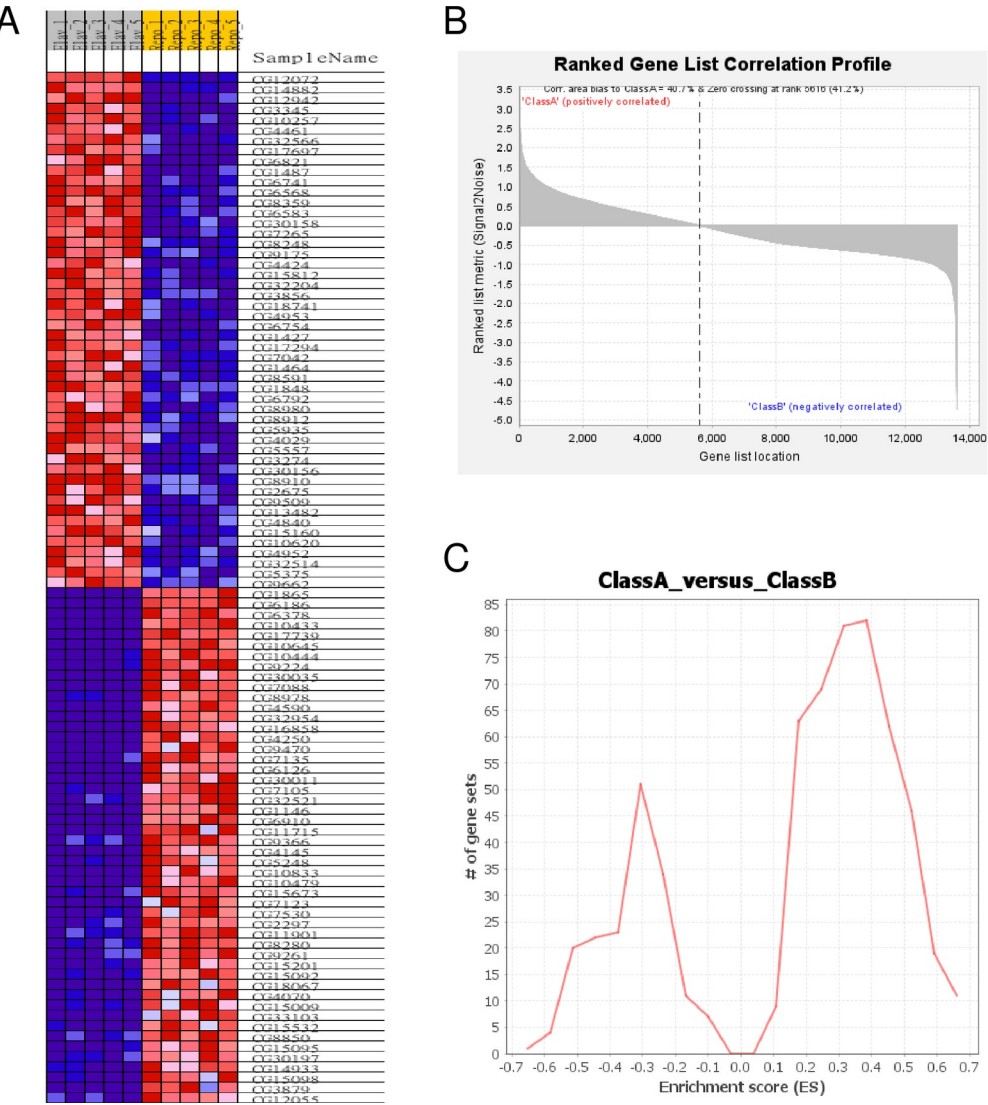

**Fig 1. Global characteristics of the ELAV vs REPO profiling analyzed with the Reactome gmt. A)** Heat Map of the top 50 features for each phenotype. **B)** Ranked Gene List Correlation Profile. **C)** Global enrichment scores across gene sets (ES histogram).

## Discussion

This study has generated gene matrix transposed files for Drosophila gene set enrichment analysis based on curated pathway databases Reactome and KEGG, containing 1450 and 131 gene sets, respectively. We validated their feasibility with GSEA 4.1.0 software by running analyses on a publically available profiling dataset of brain neurons and glia from Drosophila. Both gene matrixes separated the two groups well and highlighted the typical pathways.

Neurons are electrically excitable cells communicating with other cells through synapses by releasing or receiving neurotransmitters and dynamically modulating the density and composition of membrane receptors. In mature neurons, mRNA splicing controls ion channels, exocytosis apparatus, and neurotransmitter recycling [12]. Protein synthesis at the synapse requires transport of mature transcripts [13], while exocytosis requires RAC1 GTPase [14],

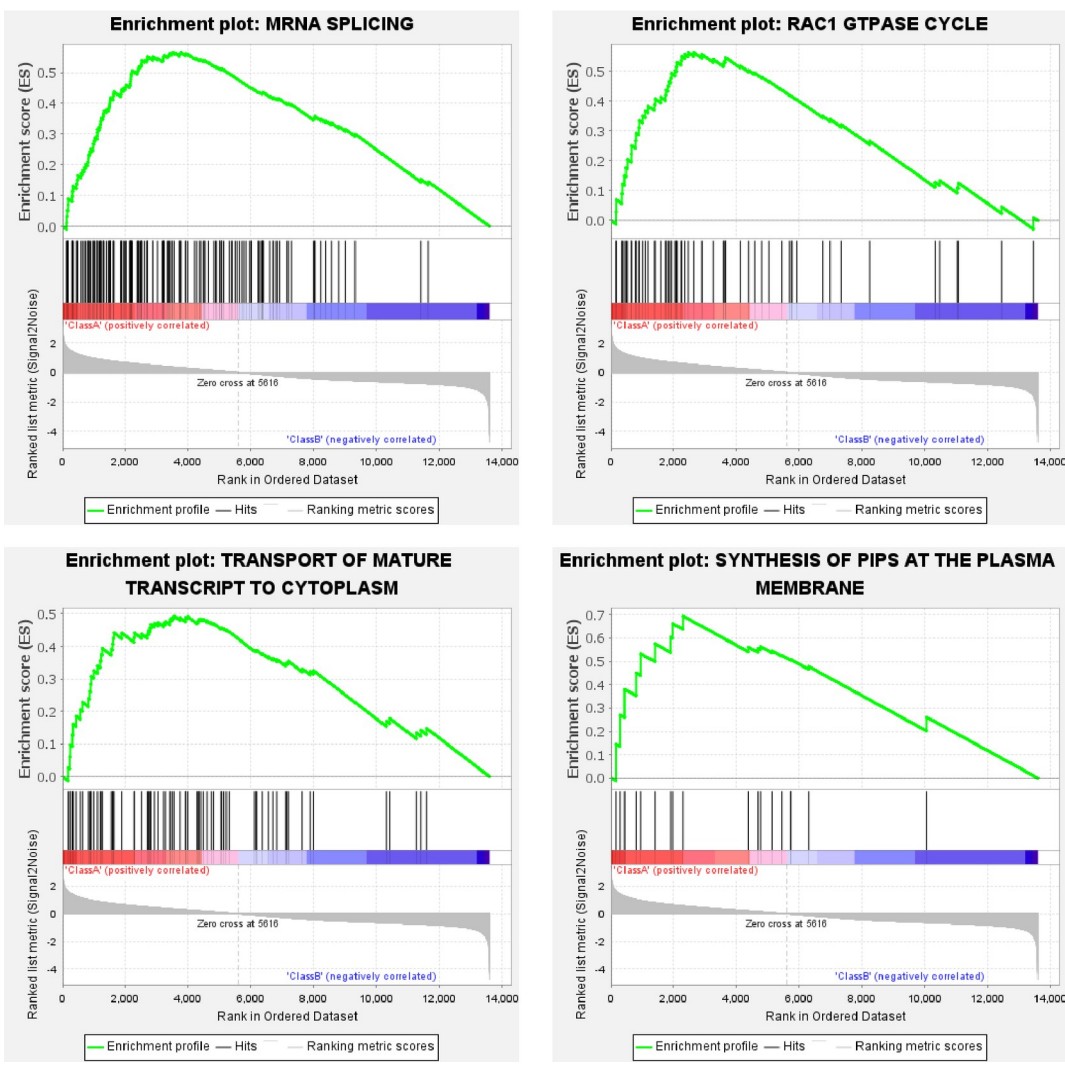

**Fig 2. Representative Reactome enrichment plots of the ELAV group.** For each plot, the upper part shows the enrichment score; while the lower part shows ranked list metric of the gene set. The middle part shows the ranked gene list, with red meaning upregulation, blue for downregulation, and black vertical line for the genes of the set.

and trafficking of synaptic vesicle membranes during the exocytic-endocytic cycle requires phosphoinositides [15]. These features are highlighted by the representative enrichment plots of neurons (Fig 2). In addition to confirming the spliceosome and endocytosis pathway, the enrichment plots (Fig 4) also highlight WNT signaling, which regulates presynaptic assembly and neurotransmitter release [16], and mTOR signaling, which participates in the dendritic spine and synapse formation [17].

Glia support the neurons in the central nervous system by forming myelin to protect and insulate neurons, by supplying nutrients and oxygen to neurons, and by destroying pathogens and removing neuronal debris [18]. The glia-neuron lactate shuttle fuels neural activity and reduces the glucose-stimulated ROS burden of neurons [19]. Meanwhile, neural lactate consumption fuels lipid production and shuttles them back to glia through ABC transports [19]. The representative enrichment plots of glia (Figs 3 and 5) highlight glutathione metabolism, which protects neurons from oxidative stress [20], and highlights also the ABC transporters,

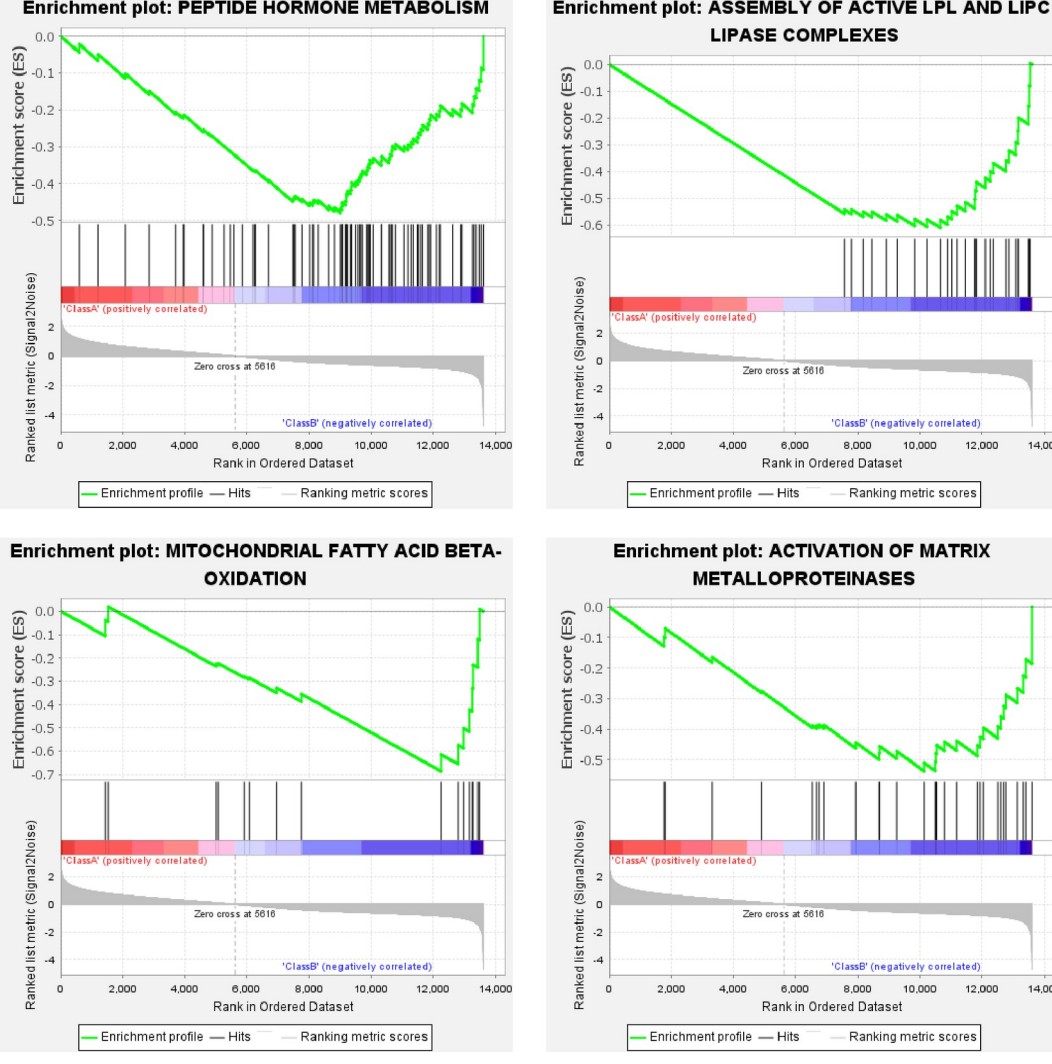

**Fig 3. Representative Reactome enrichment plots of the REPO group.** For each plot, the upper part shows the enrichment score; while the lower part shows ranked list metric of the gene set. The middle part shows the ranked gene list, with red meaning upregulation, blue for downregulation, and black vertical line for the genes of the set.

fatty acid degradation, and lipase complexes. Moreover, the glial cell also serves as a reservoir of neurotransmitters or active ligands. Among them, D-serine functions as a coagonist to NMDA receptors and controls synaptic memory [21]. Furthermore, glia induce the activity of matrix metalloproteinase 3 & 9 [22], which involves proteolysis responding to neural debris. These features are also highlighted in Fig 5.

Besides the typical pathways, the complete lists of enriched gene sets (S5–S8 Files) are resources to discover potential novel pathways in neurons or glia. For example, the gene set of "SRP-dependent cotranslational protein targeting to membrane" was enriched in glia (S6 File), but its role in glial function has not been elucidated yet.

This study may benefit future profiling studies, including GeneChip microarray and NGS, of Drosophila by enabling pathway-targeting gene set enrichment analysis. The generated Reactome and KEGG gene matrix files are compatible with GSEA 4.1.0 software. However, their compatibility with older versions has not been tested. Moreover, before utilizing the gene

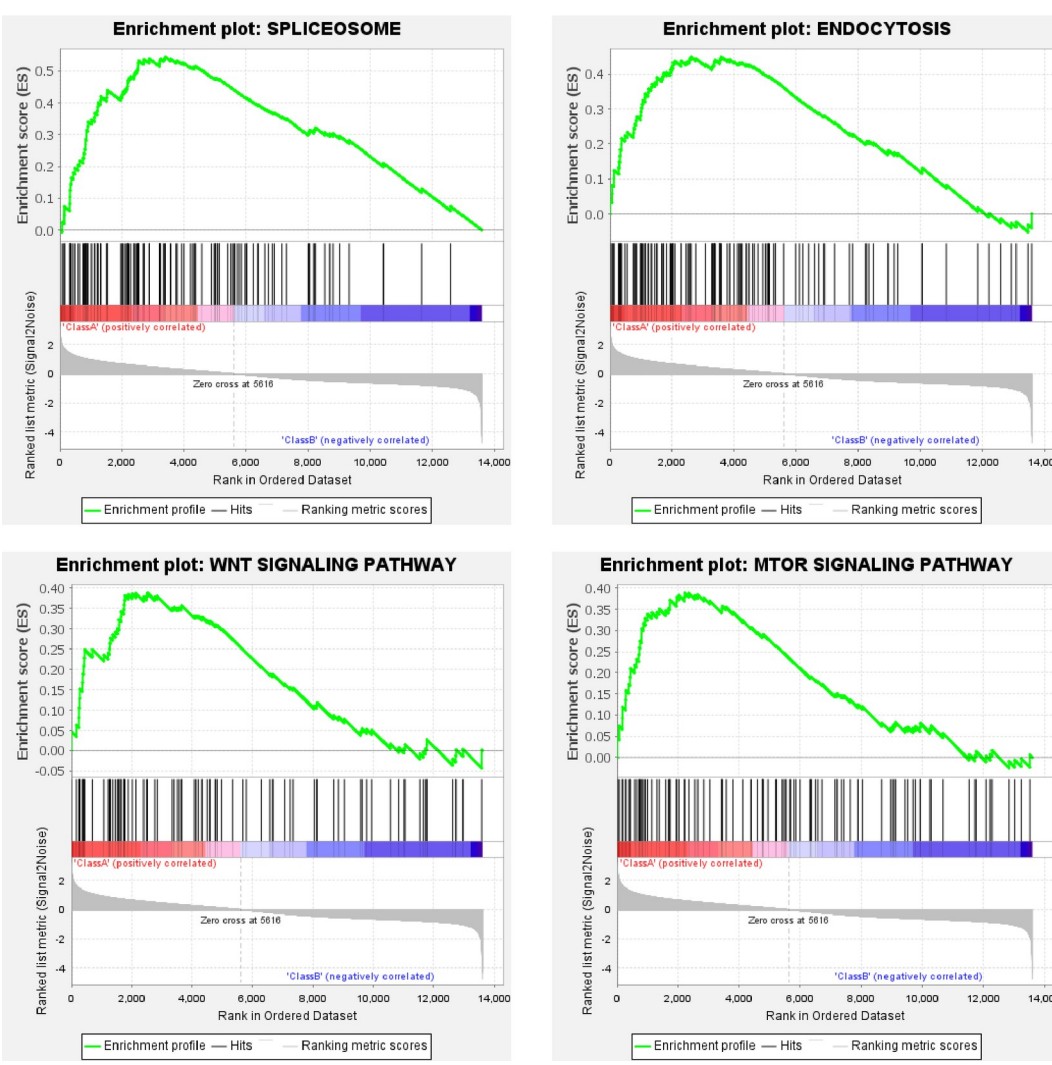

**Fig 4. Representative KEGG enrichment plots of the ELAV group.** For each plot, the upper part shows the enrichment score; while the lower part shows ranked list metric of the gene set. The middle part shows the ranked gene list, with red meaning upregulation, blue for downregulation, and black vertical line for the genes of the set.

matrix files, one has to convert the gene symbols or probe IDs in your profiling dataset into Flybase symbols (CG_ID) in advance. FlyBase ID Converter (https://www.biotools.fr/drosophila/fbgn_converter) or Gene ID conversion Tool of NIH [23] (https://david.ncifcrf.gov/conversion.jsp) may help batch ID conversion.

## Conclusion

To facilitate the pathway-targeting gene set enrichment analysis for Drosophila, we generated and validated the feasibility of Reactome and KEGG gene matrix files, which may benefit future profiling studies using Drosophila. The gene sets are available in the supplementary files or on https://github.com/JackCheng-TW/GeneMatrix. Furthermore, we have clarified the exact download specifications, ID conversion, and gmt file generation procedure in the method section so that any researcher could generate his/her gene matrix files according to the latest KEGG and Reactome databases.

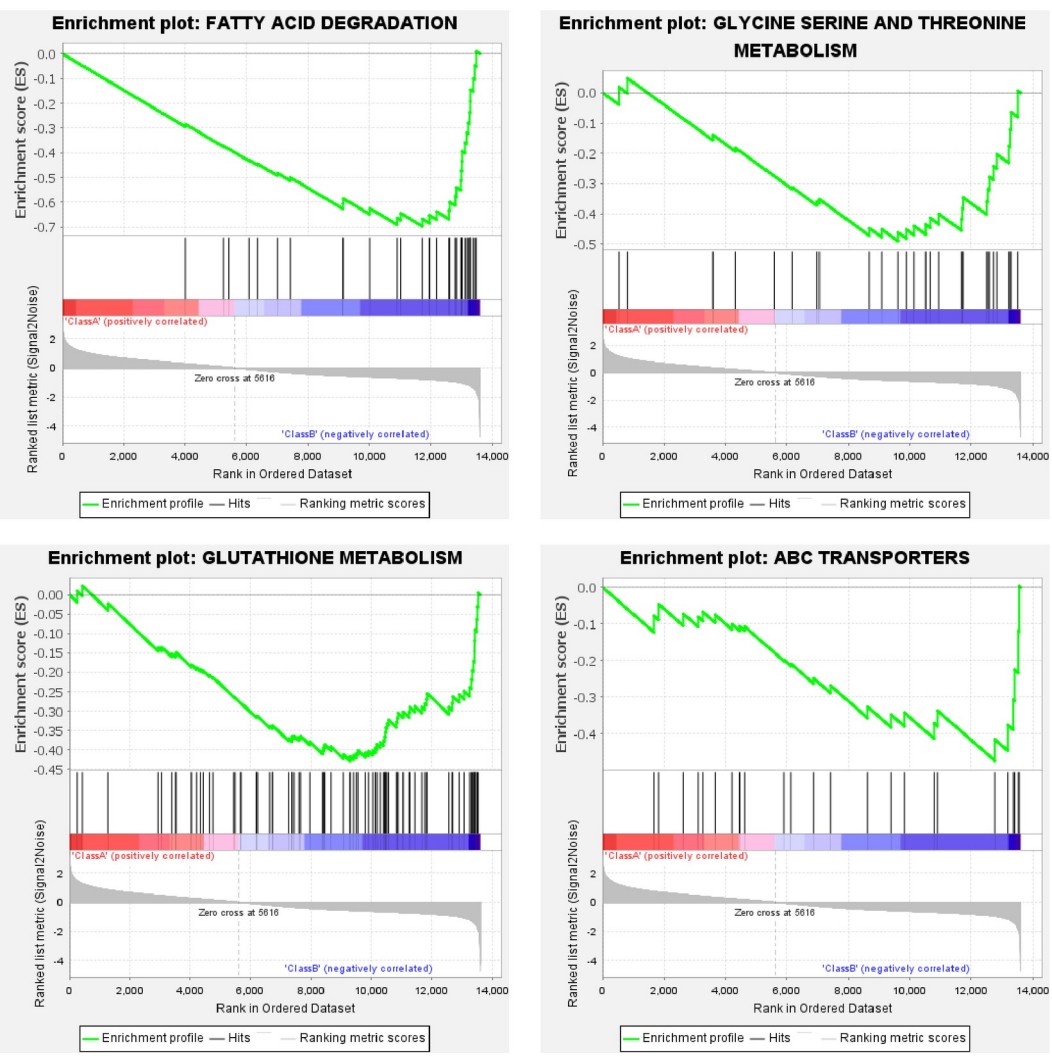

**Fig 5. Representative KEGG enrichment plots of the REPO group.** For each plot, the upper part shows the enrichment score; while the lower part shows ranked list metric of the gene set. The middle part shows the ranked gene list, with red meaning upregulation, blue for downregulation, and black vertical line for the genes of the set.

## Supporting information

**S1 File.** *Drosophila* **Reactome.**
(GMT)

**S2 File.** *Drosophila* **KEGG.**
(GMT)

**S3 File. Processed profiling of GSE45344.**
(TSV)

**S4 File. Probe ID to Flybase symbols (CG_ID) conversion table.**
(TSV)

**S5 File. Detailed Reactome enrichment results for Class A (ELAV).**
(TSV)

**S6 File. Detailed Reactome enrichment results for Class B (REPO).**
(TSV)

**S7 File. Detailed KEGG enrichment results for Class A (ELAV).**
(TSV)

**S8 File. Detailed KEGG enrichment results for Class B (REPO).**
(TSV)

**S1 Fig. The global enrichment scores across KEGG gene sets.**
(PDF)

## Author Contributions

**Conceptualization:** Hsin-Ping Liu, Wei-Yong Lin.

**Investigation:** Jack Cheng, Lee-Fen Hsu, Ying-Hsu Juan.

**Methodology:** Jack Cheng, Lee-Fen Hsu, Ying-Hsu Juan.

**Supervision:** Hsin-Ping Liu, Wei-Yong Lin.

**Writing – original draft:** Jack Cheng, Lee-Fen Hsu, Ying-Hsu Juan.

**Writing – review & editing:** Hsin-Ping Liu, Wei-Yong Lin.

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
