## [Decision Letter · Decision Letter 0]

13 Aug 2021

PONE-D-21-18414

Pathway-targeting Gene Matrix for Drosophila Gene Set Enrichment Analysis

PLOS ONE

Dear Dr. Lin,

Thank you for submitting your manuscript to PLOS ONE. After careful consideration, we feel that it has merit but does not fully meet PLOS ONE’s publication criteria as it currently stands. Therefore, we invite you to submit a revised version of the manuscript that addresses the points raised during the review process.

You will see that while the reviewers are persuaded of the importance of your GSEA datasets and validation study, there are several points highlighted that require clarification before acceptance. In particular, PLOS ONE requires methods to be described in sufficient detail for another researcher to reproduce the experiments described, so further details of the gene set generation, as suggested by reviewer 2, are needed. I also agree that submission of these gene sets to MSigDB would increase their visibility and reuse. Reviewer 1 has also highlighted some relevant studies of Gene Ontology GSEA in Drosophila which will add to the contextualisation of your work.

We look forward to receiving your revised manuscript.

Kind regards,

Katherine James, Ph.D.

Academic Editor

PLOS ONE

“This work was supported by grants from the Ministry of Science and Technology in Taiwan (MOST107-2314-B-039-042-MY2, MOST106-2314-B-039-009-, MOST108-2320-B-039-031- MY3, MOST 109-2314-B-039-030), form Chang Gung Memorial Hospital (CMRPF6H009), and from China Medical University & Hospital (CMU109-MF-85, CMU108-MF-68, CMU108-MF-61, CMU107-S-08, DMR-109-150, DMR-106-119). The funders had no role in this study.”

Funding information should not appear in the Acknowledgments section or other areas of your manuscript. We will only publish funding information present in the Funding Statement section of the online submission form.

 “This This work was supported by grants from the Ministry of Science and Technology in Taiwan (MOST107-2314-B-039-042-MY2 to WYL, MOST106-2314-B-039-009 to WYL, MOST108-2320-B-039-031-MY3 to HPL, MOST 109-2314-B-039-030 to WYL), form Chang Gung Memorial Hospital (CMRPF6H009 to LFH), and from China Medical University & Hospital (CMU109-MF-85 to WYL, CMU108-MF-68 to WYL, CMU108-MF61 to HPL, CMU107-S-08 to WYL, DMR-109-150 to WYL, DMR-106-119 to WYL). The funders had no role in this study.”

Please include your amended statements within your cover letter; we will change the online submission form on your behalf."

Reviewers' comments:

Reviewer's Responses to Questions

**Comments to the Author**

1. Is the manuscript technically sound, and do the data support the conclusions?

Reviewer #1: Partly

Reviewer #2: Yes

2. Has the statistical analysis been performed appropriately and rigorously? 

Reviewer #1: Yes

Reviewer #2: Yes

3. Have the authors made all data underlying the findings in their manuscript fully available?

Reviewer #1: Yes

Reviewer #2: Yes

4. Is the manuscript presented in an intelligible fashion and written in standard English?

Reviewer #1: Yes

Reviewer #2: Yes

5. Review Comments to the Author

Reviewer #1: Summary

In this manuscript, the authors present two alternative ways to characterize expression in Drosophila through GSEA. They develop matrices that draw on the KEGG and REACTOME pathways as opposed to Gene Ontology datasets. They then validate their matrices using datasets from Drosophila brains consisting of either neuronal or glial cells sorted by GFP+ status. Their results suggest that their matrices can effectively identify pathways that are enriched in Drosophila cells based on expression.

Major Points

• The matrices generated are specific to KEGG and REACTOME, and do seem to be novel in their use of these databases for GSEA in Drosophila melanogaster. The abstract and introduction suggest that the innovation is the use of GSEA in melanogaster in general. There are in fact matrices that have been used in melanogaster to assign GO categories. These are just a few publications that have taken advantage of GSEA for GO terms in Drosophila. It would be helpful to cite these sources and then explain what is different in these matrices (the use of different databases) and what they will provide that is different from previous GO databases in GSEA. Below are a few papers that have used GO terms in GSEA in Drosophila melanogaster.

https://www.ncbi.nlm.nih.gov/pmc/articles/PMC6114933/

https://www.g3journal.org/content/9/12/3995#ref-64

https://genomebiology.biomedcentral.com/articles/10.1186/gb-2009-10-9-r97

• Are there any REACTOME or KEGG pathways/gene sets that would have been expected to be enriched that were not? If so, what are they, and is there an explanation for these “false negatives?”

Minor Points

• “Method” is the only section with a numeral designation.

• Citation is needed in the results section for the datasets mined for validation of the matrices

Reviewer #2: GENERAL PERCEPTION

I am in general favor of the paper. The idea is obvious, as Drosophila is one of the most widely used model organisms and the availability of associated gene sets for gene set enrichment analysis is essential. The approach is as sound as it is simple and backed by the findings, which are well discussed. Another plus is that all data used in the study are provided in the supplement.

INTRODUCTION / METHODS

The paper is very brief, which I generally favor, however, I find that some basic explanations are missing and should be added. When terms such as “GFP positive cells”, “Repo”, “Elav” and “sorted” (FACS) are first mentioned, they should be briefly explained, e.g. that elav is a gene encoding an RNA binding / splicing protein and exclusively expressed in neurons, and that repo encodes a transcription factor specifically expresssed in glia, etc. Only later in Results there is a very brief part of a sentence about this: “ELAV-GFP (representing neurons) and REPO-GFP (representing glia) sorting cells from the Drosophila brain”. The introduction concludes with a statement about the objective of the study; however, it should be elaborated more on the exact approach, which is that, after generating the gene matrix files, the expression data of two distinct fruit fly cell phenotypes, i.e. neurons and glia, are run in the GSEA software to identify enriched gene sets typical for the respective cell types, which would then support the validity of the gene matrix files.

A minor discrepancy in the introduction is that, when querying the MSigDB site, there are not 32,284 gene sets of Homo sapiens, as the authors claim, but only 28,705, while the other gene sets are of four other model organisms.

According to the authors, the objective of the study is to provide two gene matrix files, which they do. However, the exact details of how these were generated are not given, as the authors merely state "The curated pathway-gene information was retrieved from the Reactome and KEGG websites. The gene matrix files [...] were generated according to the GSEA data formats". In order to reproduce these gene matrix files, a link or exact download specifications (filters, options, etc.) should be provided and information given about what gene identifiers/symbols the downloaded data came in and how these were converted to the FlyBase annotation IDs, i.e. the CG numbers. Apparently, one needs to subscribe to KEGG to download their data: https://www.kegg.jp/kegg/download/ (?). As for Reactome Pathways it is also unclear how exactly the Drosophila specific gene sets were retrieved (whereas the pathway gene sets for Homo sapiens (HSA) are easily found at https://reactome.org/download/current/ReactomePathways.gmt.zip). It should also be mentioned how the conversion table in Supplementary File 4 was obtained or generated to convert the Microarray probe IDs of the validation data to FlyBase annotation IDs. Speaking of which, the meaning of “strangest average intensity” in is a bit unclear in the sentence “Only the record with the strangest average intensity was used for the validation, in the case of redundant intensities presented for an identical CG_ID.”. Does this mean that, in a many-to-one mapping from probe IDs to CG numbers, the most extreme value was chosen in case of multiple probe mappings to the same CG number?

The steps of how the validation data and gene matrix files are run in the GSEA software are well explained.

RESULTS

The authors report the features of the generated gene matrix files and the results of the GSEA analyses correctly, except that they go by the nominal p-value when reporting significantly enriched gene sets, whereby the GSEA documentation states that due to this value not being adjusted for gene set size and multiple hypothesis testing, it is of limited value and that the FDR (false discovery rate) should be used as well (with below 25% as significant).

The highlighted representative gene sets typical for the respective cell types appear to be coherent. A minor seemingly deviating observation is that, according to the KEGG gene matrix, “ABC transporters” are (correctly) significantly enriched in the glial profile, while according to the REACTOME gene matrix, “ABC transporter in lipid homeostasis” is, though up-regulated, not significantly so. But since these two gene sets differ in genes, this might not be a meaningful comparison.

DISCUSSION

In the Discussion part, the biological implications of the highlighted enriched gene sets on neurons and glia are well discussed and backed by literature. The perspective of using the GSEA results from expression profiles of known phenotypes to explore novel pathways is a very valid point.

CONCLUSION

In the Conclusion part, it should be mentioned how/where the two gene matrix files can be accessed. Will the fruit fly researcher always have them available in the supplement of the paper or will they be submitted to the MSigDB site? On the MSigDB website, under Browse Gene Sets, there is already a category C2 for curated gene sets, specifically CP for canonical pathway, which already includes KEGG and REACTOME gene sets, just not specifically for Drosophila:

http://www.gsea-msigdb.org/gsea/msigdb/genesets.jsp?collection=CP:REACTOME

http://www.gsea-msigdb.org/gsea/msigdb/genesets.jsp?collection=CP:KEGG

Gene sets can be submitted at genesets@broadinstitute.org

As mentioned earlier, if the exact download specifications and ID conversion and .gmt file generation procedure were provided, the researcher could generate their own gene matrix files according to the authors’ procedure. This would ensure one could always obtain the latest data from the KEGG and REACTOME databases.

LANGUAGE

The text is written in well articulated language with merely a few minor wording issues, which possibly need some rephrasing; for instance, “wildly applied” probably meant “widely applied”, or “The intensity was transformed as 2 to the power of the log2-base value.” (the “log2-base value” is 2; easier: "intensity values were log2-transformed” or “log-transformed at base 2."), or “[…] citations of more than 20 thousand.” (i.e. “more than 20,000 citations”). There are very few minor misspellings, for example, “Collaspe”, a few plural/singular issues, e.g. “The […] scores […] shows […]”, and a few extra or missing “the” articles and unnecessary capitalizations, but nothing major. On a last note, it probably suffices to say "we validated the gene matrix files" and not "we validated the feasibility of the gene matrix files".

6. PLOS authors have the option to publish the peer review history of their article (what does this mean?). If published, this will include your full peer review and any attached files.

Reviewer #1: No

Reviewer #2: **Yes: **Gerrit Bostelmann

---

## [Author Response · Author response to Decision Letter 0]

15 Sep 2021

Editor’s comments

You will see that while the reviewers are persuaded of the importance of your GSEA datasets and validation study, there are several points highlighted that require clarification before acceptance. In particular, PLOS ONE requires methods to be described in sufficient detail for another researcher to reproduce the experiments described, so further details of the gene set generation, as suggested by reviewer 2, are needed.

A: The methods are described in sufficient detail for another researcher to reproduce the results. 

I also agree that submission of these gene sets to MSigDB would increase their visibility and reuse. 

A: We have contacted MSigDB to include the gene sets. However, MSigDB currently only supports Human, Mouse, and Rat gene sets, although previously MSigDB has allowed some limited deposition of gene sets from other species. Instead, MSigDB recommended making the gene sets available somewhere publicly accessible, like through a GitHub page.

We have made the gene sets available on https://github.com/JackCheng-TW/GeneMatrix.

The following sentence is added in the conclusion. The gene sets are available in the supplementary files or on https://github.com/JackCheng-TW/GeneMatrix.

Reviewer 1 has also highlighted some relevant studies of Gene Ontology GSEA in Drosophila which will add to the contextualisation of your work.

A: The advantages and disadvantages comparing GO GSEA and KEGG/Reactome GSEA are addressed, especially in the context of the suggested relevant studies.

Reviewers' comments:

Reviewer #1: Summary

In this manuscript, the authors present two alternative ways to characterize expression in Drosophila through GSEA. They develop matrices that draw on the KEGG and REACTOME pathways as opposed to Gene Ontology datasets. They then validate their matrices using datasets from Drosophila brains consisting of either neuronal or glial cells sorted by GFP+ status. Their results suggest that their matrices can effectively identify pathways that are enriched in Drosophila cells based on expression.

Major Points

• The matrices generated are specific to KEGG and REACTOME, and do seem to be novel in their use of these databases for GSEA in Drosophila melanogaster. The abstract and introduction suggest that the innovation is the use of GSEA in melanogaster in general. There are in fact matrices that have been used in melanogaster to assign GO categories. These are just a few publications that have taken advantage of GSEA for GO terms in Drosophila. It would be helpful to cite these sources and then explain what is different in these matrices (the use of different databases) and what they will provide that is different from previous GO databases in GSEA. Below are a few papers that have used GO terms in GSEA in Drosophila melanogaster.

https://www.ncbi.nlm.nih.gov/pmc/articles/PMC6114933/

https://www.g3journal.org/content/9/12/3995#ref-64

https://genomebiology.biomedcentral.com/articles/10.1186/gb-2009-10-9-r97

A: Gene Ontology is a structured, precisely defined, controlled vocabulary for describing the roles of genes with three independent ontologies, i.e., biological process, molecular function, and cellular component (Ashburner, Michael, et al. "Gene ontology: tool for the unification of biology." Nature genetics 25.1 (2000): 25-29.). Briefly, biological process indicates the biological objective of the gene/protein; molecular function describes its biochemical activity, while cellular component refers to its subcellular distribution. Although GO allows an easy and quick understanding of the roles of a gene, however, “it describes only what is done without specifying where or when the event actually occurs.” as stated in the original GO paper (Ashburner, Michael, et al. "Gene ontology: tool for the unification of biology." Nature genetics 25.1 (2000): 25-29.). This knowledge gap is exactly what KEGG and REACTOME try to fill. Both databases provide sequential information and partnership of the reaction of the gene/protein. On the contrary, the dependence on the published/curated scientific literature largely limits the application of KEGG/REACTOME on genes of unknown function, while GO may cover this part by similarity prediction.

Thus, the choice of gene sets in GSEA is largely dependent on the purpose of the study. For example, the first two GSEA papers (PMC6114933 and 3995#ref-64) may be improved by adopting KEGG/REACTOME gene sets to provide more details of the affected pathways, while for the third paper (gb-2009-10-9-r97), GO gene sets is perfect for its goal of predicting functions of unknown genes.

• Are there any REACTOME or KEGG pathways/gene sets that would have been expected to be enriched that were not? If so, what are they, and is there an explanation for these "false negatives?"

A: There are indeed crucial pathways that remain unannotated in both databases, such as Draper-dependent glial phagocytic activity, Draper‐mediated JNK signaling, and Glial phagocytosis. These are representative pathways characterizing glial activity. Another problem is the missing critical proteins in the annotated pathway; for example, serine racemase (CG8129) is not annotated in the serine biosynthesis pathway (R-DME-977347). Pathway databases are not perfect but still represent the state-of-the-art knowledge of the research community. These "false negatives" represent the knowledge gaps for us to explore.

Minor Points

• "Method" is the only section with a numeral designation.

A: Thanks for this. The numeral designation of “Method” is removed.

• Citation is needed in the results section for the datasets mined for validation of the matrices

A: DeSalvo, Michael K., et al. "The Drosophila surface glia transcriptome: evolutionary conserved blood-brain barrier processes." Frontiers in neuroscience 8 (2014): 346.

Reviewer #2: GENERAL PERCEPTION

I am in general favor of the paper. The idea is obvious, as Drosophila is one of the most widely used model organisms and the availability of associated gene sets for gene set enrichment analysis is essential. The approach is as sound as it is simple and backed by the findings, which are well discussed. Another plus is that all data used in the study are provided in the supplement.

INTRODUCTION / METHODS

The paper is very brief, which I generally favor, however, I find that some basic explanations are missing and should be added. When terms such as "GFP positive cells", "Repo", "Elav" and "sorted" (FACS) are first mentioned, they should be briefly explained, e.g. that elav is a gene encoding an RNA binding / splicing protein and exclusively expressed in neurons, and that repo encodes a transcription factor specifically expresssed in glia, etc. Only later in Results there is a very brief part of a sentence about this: "ELAV-GFP (representing neurons) and REPO-GFP (representing glia) sorting cells from the Drosophila brain". 

A: These sentences are added. Elav is a gene encoding an RNA binding protein capable of regulating mRNA processing exclusively expressed in neurons, and Repo encodes a transcription factor specifically expressed in glia. By using GAL4-UAS reporter system, Repo-GAL4 drives the expression of UAS-GFP specifically in glia, while Elav-GAL4 drives UAS-GFP exclusively in neurons. Fluorescence activated cell sorting (FACS) is a technique to separate cells as they flow past stimulating lasers (Bonner, W. A., et al. "Fluorescence activated cell sorting." Review of Scientific Instruments 43.3 (1972): 404-409.).

The introduction concludes with a statement about the objective of the study; however, it should be elaborated more on the exact approach, which is that, after generating the gene matrix files, the expression data of two distinct fruit fly cell phenotypes, i.e. neurons and glia, are run in the GSEA software to identify enriched gene sets typical for the respective cell types, which would then support the validity of the gene matrix files.

A: This sentence is added. After generating the gene matrix files, the expression data of two distinct fruit fly cell phenotypes, i.e., neurons and glia, are run in the GSEA software to identify enriched gene sets typical for the respective cell types, which would support the validity of the gene matrix files.

A minor discrepancy in the introduction is that, when querying the MSigDB site, there are not 32,284 gene sets of Homo sapiens, as the authors claim, but only 28,705, while the other gene sets are of four other model organisms.

A: Thanks a lot. The number is modified.

According to the authors, the objective of the study is to provide two gene matrix files, which they do. However, the exact details of how these were generated are not given, as the authors merely state "The curated pathway-gene information was retrieved from the Reactome and KEGG websites. The gene matrix files [...] were generated according to the GSEA data formats". 

In order to reproduce these gene matrix files, a link or exact download specifications (filters, options, etc.) should be provided and information given about what gene identifiers/symbols the downloaded data came in and how these were converted to the FlyBase annotation IDs, i.e. the CG numbers. Apparently, one needs to subscribe to KEGG to download their data: https://www.kegg.jp/kegg/download/ (?). As for Reactome Pathways it is also unclear how exactly the Drosophila specific gene sets were retrieved (whereas the pathway gene sets for Homo sapiens (HSA) are easily found at https://reactome.org/download/current/ReactomePathways.gmt.zip). 

A: The Drosophila-specific genes of KEGG pathways were downloaded from the “KEGG Pathway Maps - Drosophila melanogaster (fruit fly)” with the website https://www.genome.jp/brite/query=00190&htext=br08901.keg&option=-a&node_proc=br08901_org&proc_enabled=dme&panel=collapse. By clicking each “tringle symbol” of mother categories, sub-categories will expand. By clicking the number preceding each sub-category, e.g., 00010 of Glycolysis / Gluconeogenesis, it will bring you to the map of the specific pathway (https://www.genome.jp/kegg-bin/show_pathway?dme00010). Further clicking the title of pathway map on the upper left corner will finally bring you to the detail page of that pathway (https://www.genome.jp/entry/dme00010). At the upper right corner of the page, an “all links” box contains the “KEGG GENES” list. Repeat the process to exhaust the Drosophila KEGG pathways.

The Drosophila-specific genes of REACTOME pathways were downloaded from the (https://reactome.org/PathwayBrowser/#/R-DME-XXXXXXX, where XXXXXXX denotes for seven digits of a specific pathway, e.g., 9612973 for autophagy). There are three panels on the page. The left panel shows the hierarchy of Drosophila pathways in REACTOME, while at the right lower panel, by clicking the tab “Molecules”, then the “protein” link, the gene/protein list is available. Repeat the process to exhaust the Drosophila REACTOME pathways on the left panel.

A gmt file is a tab-separated plain text, and each row describes one gene set. In each row, the first column contains the name of the gene set, while the second column contains additional details, e.g., KEGG ID of the pathway (gene set). The gene set members, i.e., FlyBase IDs in this study, are listed from the third column of the row, one gene in one column. Thus, once the gene list of pathways is available, the gmt file can be generated by locating the pathway elements into corresponding cells with any plain text editor or Microsoft Excel. After saving the file as a tab-separated plain text, modify the filename extension, i.e., *.txt, to *.gmt in the file browser.

It should also be mentioned how the conversion table in Supplementary File 4 was obtained or generated to convert the Microarray probe IDs of the validation data to FlyBase annotation IDs. 

A: The annotation of Microarray probe IDs is available from the “SOFT formatted family file” at https://ftp.ncbi.nlm.nih.gov/geo/series/GSE45nnn/GSE45344/soft/. Specifically, from line 110 of the file, the 1st column is the Microarray probe ID, and the 3rd column is the FlyBase annotation ID. However, the context must be “cleared” to extract the correct FlyBase ID, e.g., for “CG16844-RA”, the “-RA” must be trimmed to get the correct ID “CG16844”. In the case that the dataset of your interest does not provide the probe ID annotation, you may try the gene ID conversion tool on https://david.abcc.ncifcrf.gov/conversion.jsp or from the website of the gene chip manufacturer.

Speaking of which, the meaning of "strangest average intensity" in is a bit unclear in the sentence "Only the record with the strangest average intensity was used for the validation, in the case of redundant intensities presented for an identical CG_ID.". Does this mean that, in a many-to-one mapping from probe IDs to CG numbers, the most extreme value was chosen in case of multiple probe mappings to the same CG number?

A: Yes, the sentence is added. The highest value was chosen in case of multiple probe mappings to the same CG ID.

The steps of how the validation data and gene matrix files are run in the GSEA software are well explained.

A: Thanks.

RESULTS

The authors report the features of the generated gene matrix files and the results of the GSEA analyses correctly, except that they go by the nominal p-value when reporting significantly enriched gene sets, whereby the GSEA documentation states that due to this value not being adjusted for gene set size and multiple hypothesis testing, it is of limited value and that the FDR (false discovery rate) should be used as well (with below 25% as significant).

A: We included the FDR 25% criterion, and therefore the “significant” gene sets now have to meet both FDR and nominal p-value. (Fortunately,) in this study, the gene sets with significant nominal p-value also meet the FDR 25% criterion (Supp File 5, 6, 7, 8). Therefore, the list of significant gene sets does not change due to the inclusion of FDR criterion. The description is added “under the criterion of false discovery rate (FDR) below 25%” when applicable.

The highlighted representative gene sets typical for the respective cell types appear to be coherent. A minor seemingly deviating observation is that, according to the KEGG gene matrix, "ABC transporters" are (correctly) significantly enriched in the glial profile, while according to the REACTOME gene matrix, "ABC transporter in lipid homeostasis" is, though up-regulated, not significantly so. But since these two gene sets differ in genes, this might not be a meaningful comparison.

A: ABC transporters (KEGG dme02010) couple ATP hydrolysis to active transport of a wide variety of substrates such as ions, sugars, lipids, sterols, peptides, proteins. While ABC transporter in lipid homeostasis (Reactome R-DME-1369062) is only a part of the “ABC transporters”.

DISCUSSION

In the Discussion part, the biological implications of the highlighted enriched gene sets on neurons and glia are well discussed and backed by literature. The perspective of using the GSEA results from expression profiles of known phenotypes to explore novel pathways is a very valid point.

A:Thanks a lot.

CONCLUSION

In the Conclusion part, it should be mentioned how/where the two gene matrix files can be accessed. Will the fruit fly researcher always have them available in the supplement of the paper or will they be submitted to the MSigDB site? On the MSigDB website, under Browse Gene Sets, there is already a category C2 for curated gene sets, specifically CP for canonical pathway, which already includes KEGG and REACTOME gene sets, just not specifically for Drosophila:

http://www.gsea-msigdb.org/gsea/msigdb/genesets.jsp?collection=CP:REACTOME

http://www.gsea-msigdb.org/gsea/msigdb/genesets.jsp?collection=CP:KEGG

Gene sets can be submitted at genesets@broadinstitute.org

A: We have contacted MSigDB to include the gene sets. However, MSigDB currently only supports Human, Mouse, and Rat gene sets, although previously MSigDB has allowed some limited deposition of gene sets from other species. Instead, MSigDB recommended making the gene sets available somewhere publicly accessible, like through a GitHub page.

We have made the gene sets available on https://github.com/JackCheng-TW/GeneMatrix.

The following sentence is added in the conclusion. The gene sets are available in the supplementary files or on https://github.com/JackCheng-TW/GeneMatrix.

As mentioned earlier, if the exact download specifications and ID conversion and .gmt file generation procedure were provided, the researcher could generate their own gene matrix files according to the authors' procedure. This would ensure one could always obtain the latest data from the KEGG and REACTOME databases.

A: The following sentence is added in the conclusion. We have clarified the exact download specifications, ID conversion, and gmt file generation procedure in the method section so that any researcher could generate his/her gene matrix files according to the latest KEGG and REACTOME databases.

LANGUAGE

The text is written in well articulated language with merely a few minor wording issues, which possibly need some rephrasing; for instance, "wildly applied" probably meant "widely applied", or "The intensity was transformed as 2 to the power of the log2-base value." (the "log2-base value" is 2; easier: "intensity values were log2-transformed" or "log-transformed at base 2."), or "[...] citations of more than 20 thousand." (i.e. "more than 20,000 citations"). There are very few minor misspellings, for example, "Collaspe", a few plural/singular issues, e.g. "The [...] scores [...] shows [...]", and a few extra or missing "the" articles and unnecessary capitalizations, but nothing major. On a last note, it probably suffices to say "we validated the gene matrix files" and not "we validated the feasibility of the gene matrix files".

A: Thank you very much. Modified as suggested.

---

## [Decision Letter · Decision Letter 1]

15 Oct 2021

Pathway-targeting Gene Matrix for Drosophila Gene Set Enrichment Analysis

PONE-D-21-18414R1

Dear Dr. Lin,

We’re pleased to inform you that your manuscript has been judged scientifically suitable for publication and will be formally accepted for publication once it meets all outstanding technical requirements.

Kind regards,

Katherine James, Ph.D.

Academic Editor

PLOS ONE

Additional Editor Comments (optional):

Reviewers' comments:

Reviewer's Responses to Questions

**Comments to the Author**

1. If the authors have adequately addressed your comments raised in a previous round of review and you feel that this manuscript is now acceptable for publication, you may indicate that here to bypass the “Comments to the Author” section, enter your conflict of interest statement in the “Confidential to Editor” section, and submit your "Accept" recommendation.

Reviewer #1: All comments have been addressed

Reviewer #2: All comments have been addressed

2. Is the manuscript technically sound, and do the data support the conclusions?

Reviewer #1: Yes

Reviewer #2: Yes

3. Has the statistical analysis been performed appropriately and rigorously? 

Reviewer #1: Yes

Reviewer #2: Yes

4. Have the authors made all data underlying the findings in their manuscript fully available?

Reviewer #1: Yes

Reviewer #2: Yes

5. Is the manuscript presented in an intelligible fashion and written in standard English?

Reviewer #1: Yes

Reviewer #2: Yes

6. Review Comments to the Author

Reviewer #1: The authors have satisfactorily addressed all of my concerns. They have included explanation of how their analysis differs from previous analyses, giving context to the audience.

Reviewer #2: (No Response)

7. PLOS authors have the option to publish the peer review history of their article (what does this mean?). If published, this will include your full peer review and any attached files.

Reviewer #1: No

Reviewer #2: **Yes: **Gerrit Bostelmann

---

## [Editor Report · Acceptance letter]

19 Oct 2021

PONE-D-21-18414R1 

Pathway-targeting Gene Matrix for Drosophila Gene Set Enrichment Analysis 

Dear Dr. Lin:

I'm pleased to inform you that your manuscript has been deemed suitable for publication in PLOS ONE. Congratulations! Your manuscript is now with our production department. 

Kind regards, 

on behalf of

Dr. Katherine James 

Academic Editor

PLOS ONE